# Chronology of COVID-19 Symptoms in Very Old Patients: Study of a Hospital Outbreak

**DOI:** 10.3390/jcm10132962

**Published:** 2021-06-30

**Authors:** Carmelo Lafuente-Lafuente, Quoc Duy Nghiem, Héloïse Keravec, Sihem Oukbir-Ferrag, Maurizio Magri, Bruno Oquendo, Cristiano Donadio, Antonio Rainone, Joël Belmin

**Affiliations:** 1Service de Gériatrie à Orientation Cardiologique et Neurologique, AP-HP, Sorbonne Université, Hôpitaux Universitaires Pitie-Salpêtrière-Charles Foix, F-94200 Ivry-sur-Seine, France; quocduy.nghiem@aphp.fr (Q.D.N.); sihem.ferrag@aphp.fr (S.O.-F.); maurizio.magri@aphp.fr (M.M.); bruno.oquendo@aphp.fr (B.O.); cristiano.donadio@aphp.fr (C.D.); j.belmin@aphp.fr (J.B.); 2Clinical Epidemiology and Ageing (CEpiA) Unit, INSERM U955, Institut Mondor de Recherche Biomédicale, F-9410 Créteil, France; 3Unité de Gériatrie Aigue, AP-HP, Sorbonne Université, Hôpital Rothschild, F-75012 Paris, France; heloisekeravec@gmail.com; 4Service de Soins de Suite et de Réadaptation Gériatrique, AP-HP, Sorbonne Université, Hôpitaux Universitaires Pitie-Salpêtrière-Charles Foix, F-94200 Ivry-sur-Seine, France; antonio.rainone@aphp.fr

**Keywords:** COVID-19, older patients, atypical symptoms, early diagnosis, temporal pattern

## Abstract

Background: We wanted to better understand the frequency and temporal distribution of symptoms of COVID-19 in very old patients, which are currently not well defined. Methods: In an observational, descriptive study, we followed all patients being at three geriatric convalescence and rehabilitation units when a COVID-19 outbreak emerged in those units in March 2020. For those who developed the disease, we recorded any new symptom occurring at diagnosis, in the previous 14 and the following 21 days. A group of SARS-Cov-2-negative patients served as controls. Results: Sixty-nine of the 176 inpatients (mean age: 86 years) were infected by SARS-Cov-2 during the outbreak. At the moment of diagnosis, a majority of patients had fever (71.0%), malaise-asthenia (24.6%), or respiratory symptoms (66.7%). However, 48 patients (69.6% of all SARS-Cov-2 positive patients) also presented, usually several days before, other symptoms: (a) gastrointestinal symptoms (39.1% of all patients, median onset eight days before diagnosis, IQR −9 to +3 days); (b) neurological symptoms (30.4% of all patients, median onset five days before diagnosis, IQR −11 to −3 days), notably delirium (24.6%); and (c) other symptoms, like falls and unexplained decompensation of chronic conditions (29.0% of all patients, median onset four days before diagnosis, IQR −10 to 0). None of those symptoms were observed in similar proportion in 25 control SARS-CoV-2-negative patients, hospitalized during the same period. Conclusions: Diarrhea, nausea-vomiting, delirium, falls, and unexplained decompensation of chronic conditions were the first symptoms of COVID-19 in a majority of older patients in this cohort, preceding typical symptoms by several days. Recognizing those early symptoms could hasten the diagnosis of COVID-19 in this population.

## 1. Introduction

Older patients constitute a substantial proportion of all COVID-19 patients, develop severe disease more frequently, and cumulate a majority of the deaths caused by COVID-19, especially if very old or frail [1,2,3,4,5]. The extent to which the clinical presentation of COVID-19 may be different in older patients is not well known. Published studies to date have reached inconsistent results. Some studies, including a small series, a retrospective cohort, and a national cross-sectional study, have found that atypical symptoms, like diarrhea, falls, and delirium, were more frequent in older patients [6,7], especially in frail ones [8]. This was also the impression of some clinicians who had treated older COVID-19 patients [9]. However, other series and large studies found either no difference compared with younger patients [10,11], or found, as the only difference, that typical symptoms (fever, malaise, respiratory) were less frequent in older patients [12,13,14]. Solving those discordances is important, as not recognizing early symptoms of COVID-19 might lead to delayed diagnosis and treatment, as well as allowing more time to further spread the infection between contacts.

In mid-March 2020, as the COVID-19 pandemic wave reached the Paris area, France, outbreaks of COVID-19 developed in several geriatric rehabilitation units, affecting patients who had been initially hospitalized for other causes. That provided us a unique opportunity to observe and follow the clinical characteristics of COVID-19 as they developed in those patients. Therefore, we decided to systematically record any symptom that appeared, and its time of onset, in all patients who acquired COVID-19, as well as in a control group of patients who tested negative for SARS-CoV-2, to better define which were the most frequent symptoms of COVID-19 in older people, and to determine if those symptoms followed any temporal pattern.

## 2. Methods

This was an observational study, with a retrospective cohort, conducted in three geriatric rehabilitation and convalescence units, totaling 176 beds, located at two hospitals, in the Paris area, France.

Patients included were older adults (>75 years old) who were hospitalized at the participating units and developed SARS-CoV-2 infection during their stay. Patients had been initially hospitalized for a variety of medical conditions and had been transferred from acute care because they needed rehabilitation or more time to recover. After the first Covid-19 cases appeared, all patients who had been in contact with them were systematically tested, whether symptomatic or not, as well as any patient subsequently presenting any new or unusual symptom, of any severity, even very mild. All diagnostic were made by detecting SARS-CoV-2 specific ARN (E gen and/or Orf region) in nasopharyngeal swab samples, using an automated PCR Cobas^®^ 6800 System assay (Roche Ltd., Basel, Switzerland).

For each COVID-19 patient, we retrieved and recorded, using a predefined form, any new clinical sign or symptom that was present at the time of diagnosis, have been noticed in the previous 14 days or appeared in the following 21 days, as well as the moment it appeared. Data were retrieved prospectively from the moment of diagnostic for some patients, but retrospectively from their medical records for others. In addition, in order to differentiate those signs and symptoms from other acute complications frequent in geriatric patients (e.g., other infections), we also retrieved symptoms, in the same way, in all patients who were in one of the participating units at the beginning of the outbreak but who tested negative for SARS-CoV-2. Variables in both groups (COVID-19 and controls) were compared using Fisher exact test (for frequencies) or a Wilcoxon rank sum test (when numerical), employing STATA (v13, StataCorp LP, Texas, TX, USA). All tests were two-tailed and a *p* < 0.05 was considered significant.

All patients received written information about the use of their personal medical data for research purposes and could refuse at any moment to participate in the study. All data was anonymized. The French *Commission Nationale Informatique et Libertés (CNIL)* authorized the use of medical data for this study (Declaration number: 2218210v0). The study was conducted in full accordance with the Declaration of Helsinki. We followed the STROBE statement for improving the reporting of observational studies while writing this manuscript [15]. A completed STROBE checklist is available from the authors upon request.

## 3. Results

After the first PCR-confirmed COVID-19 case, on March 19, 2020, the number of cases quickly increased and reached a peak eight days later. An enquiry found no preceding suspected cases in patients or in hospital staff, but at least two confirmed Covid-19 cases in relatives who had recently visited their loved ones (visits were later forbidden, in 12 March). We concluded that the disease had probably been introduced by visiting relatives, then involuntarily disseminated among in-patients by caregivers and patient-to-patient contact. A separate division was created to host and isolate all COVID-19 patients and after 20 April no new case appeared for the following four weeks, when this study was closed.

In the end, 69 of the 176 inpatients (39.2%) acquired a confirmed SARS-CoV-2 infection during the study period (Figure 1). Mean age was 86 (SD ±6.4) years. Six patients (8.7%) were asymptomatic at all times. They were tested, and thus diagnosed, because have been in contact with known cases. The characteristics of patients and the distribution of symptoms are shown in Table 1.

The most frequent symptoms were fever, associated or not with general symptoms (71.0% of patients), and respiratory symptoms (66.7% of patients). Eighteen patients (26.1%) had oxygen desaturation at least at one moment, but only four patients (5.8%) developed overt respiratory failure. Fever, general and respiratory symptoms usually appeared at the same time, as shown in Figure 2, and prompted the diagnosis of COVID-19 in most patients. Overall, 57 patients (82.6%) had one (or several) of those symptoms. 

In the horizontal axis, the 0 point corresponds to the moment COVID-19 was diagnosed. Negative numbers denote an onset of symptoms prior to COVID-19 diagnosis, positive numbers an onset of symptom afterwards.

In addition, 48 patients (69.6%) presented symptoms other than fever, general or respiratory. Twenty seven patients (39.1%) had gastrointestinal symptoms, diarrhea being by large the most frequent (34.8% of all patients). Twenty one (30.4%) presented neurological symptoms, particularly the appearance of unexplained delirium (24.6% of all patients) and less frequently behavioural disturbances (7.2%). Finally, some patients had other, diverse symptoms, such as falls (15.6% of patients) and sudden decompensation of previously stable chronic diseases, predominantly of heart failure (13.0%). None of those symptoms happened with similar frequency in control, COVID-19-negative patients, during the study period.

A distinctive temporal pattern appeared when analyzing the time of onset of each symptom, as shown in Table 1 and Figure 2. Gastrointestinal, neurological, and other symptoms tended to appear early in the course of disease, preceding fever and respiratory symptoms by several days: neurological symptoms appeared a median of eight days before (interquartile range (IQR) −11 to −3); gastrointestinal symptoms a median of five days before (IQR −9 days before to +3 days after); and other symptoms (falls, heart failure) a median of four days before (IQR −10 to 0).

## 4. Discussion

In this study, we found that a great proportion of older patients had other symptoms than fever and respiratory. Almost 40% also had gastrointestinal symptoms and about 30% presented neurological or other symptoms. Diarrhea, occurrence of unexplained delirium and falls were respectively the leading symptom in each group. These symptoms are relatively frequent in older patients, as they can be caused by a variety of conditions, but none of them happened with similar frequency in the control group during the same period. 

The main finding of our study, however, is a definite temporal pattern: gastrointestinal, neurological, and other atypical symptoms had a clear tendency to appear early in the course of the infection. They preceded by several days the appearance of fever and respiratory symptoms, which, in most cases, were the symptoms that finally led us to diagnosing COVID-19. Luo et al. also found that gastrointestinal symptoms were often the first symptoms appearing [16]. No other study, however, has reported a similar temporal distribution of symptoms, to the best of our knowledge. Fever, respiratory symptoms, general symptoms (fatigue, myalgia, asthenia …), and anosmia are reported in virtually all studies as the first symptoms appearing in COVID-19 [1,2,3,4,5,6,7,8,9,10,11,12,13,17]. Larsen et al. specifically studied the order of symptoms onset in a dataset of more than 55,000 COVID-19 patients from China and found that the most likely order of occurrence of symptoms was fever, cough, nausea/vomiting, and diarrhea [18]. Most patients in the dataset they employed were young adults, however, and a separate analysis of older adults was not conducted.

Additionally, our findings confirm those from other studies indicating that atypical symptoms of COVID-19 are more frequent in older patients [6,7]. Gastrointestinal and neurologic syndromes have been well described in COVID-19 patients but less frequently. Gastrointestinal symptoms have been found in 11–19% of all patients [2,16,19,20]. Neurological symptoms have been reported less consistently and in just between 3% and 9% of patients, if we exclude unspecific symptoms like headache and dizziness [1,2,21,22,23]. Published studies in older patients have found inconsistent results regarding the type and frequency of COVID-19 symptoms [5,6,7,8,9,10,11,12,13]. These discrepancies are probably due to variations in the characteristics of included patients, setting of the study and methods employed. The mean age of included patients, for instance, varied largely, ranging from 67 to 86.5 years. Another important factor is frailty, which has been associated COVID-19 severity and variations in clinical features [3,4,7]. Patients in our study were very old (mean 86 years) and, although we did not formally assess frailty, most of them had chronic conditions and limitations in their autonomy that would qualify them as frail. In any case, several published studies have also found that atypical presentations were more frequent in older persons [5,6,11,12,13]. None of those studies, however, analyzed the order of symptoms onset. In most of them symptoms were only assessed at inclusion.

The fact that the chronology of symptoms may be very different in very old persons has important clinical implications, as recognizing those first, less typical symptoms might allow to diagnose COVID-19 several days earlier in many of those patients. An early diagnosis is important, not only for starting proper treatment of affected patients, but also in order to rapidly implement control measures to limit the virus spreading. The latter is essential in these settings, at hospitals and nursing homes, where many older persons stay together, to limit the number of persons contaminated. In this regard, it must be noted that, as in other studies in older patients [24,25], some of our patients showed active nasopharyngeal excretion of SARS-CoV-2, as confirmed by PCR, but were completely asymptomatic, even though they were very old and fragile. 

Our work has clear limitations: the number of patients included is limited, they were very old and most of them had diverse chronic conditions, part of the data was retrieved retrospectively, and older patients were not directly compared with younger ones. Consequently, it remains uncertain whether our findings could be applied to the general population of frail or very old patients. Further studies would be needed to confirm its generalizability. Assessing frailty status in those studies would be important. As strengths, including all patients involved in an outbreak allowed us to study patients from the whole spectrum of severity of the disease, as well as methodically retrieve any symptom appearing during the entire clinical course of the disease. 

In conclusion, in this cohort of very old patients with COVID-19, the first symptoms of COVID-19 in a majority of patients (more than two thirds) were gastrointestinal, neurological, and other atypical symptoms, particularly delirium, falls, and sudden decompensation of chronic conditions. Those symptoms showed a definite temporal pattern, usually appearing several days before fever and respiratory symptoms. Recognizing this temporal pattern could hasten the diagnosis of COVID-19 in older patients by several days. 

More research remains necessary to know if COVID-19 symptoms follow a similar pattern in other groups of older patients. 

## Figures and Tables

**Figure 1 jcm-10-02962-f001:**
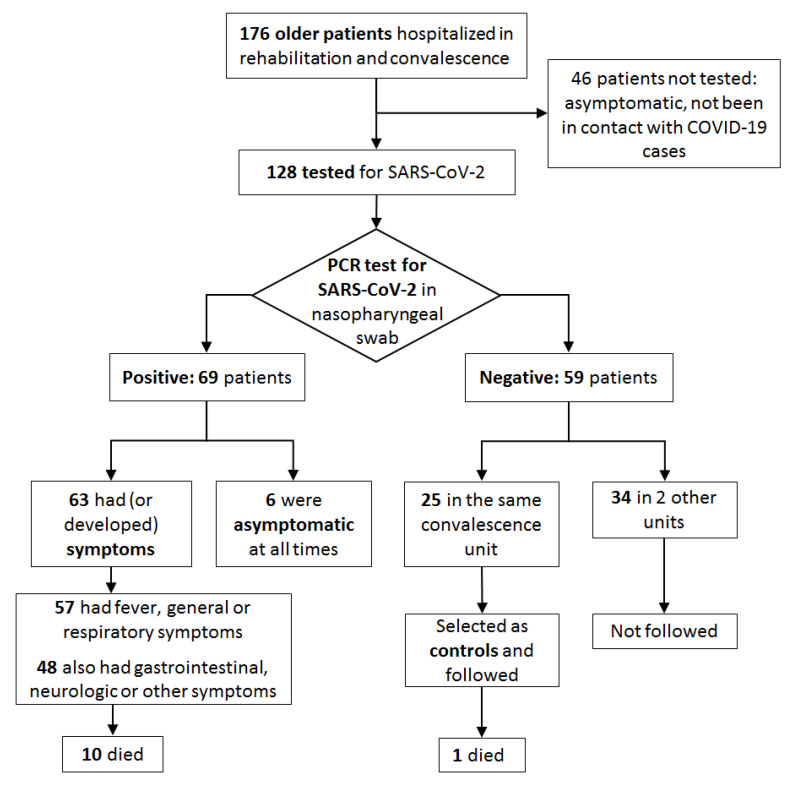
Inclusion of patients.

**Figure 2 jcm-10-02962-f002:**
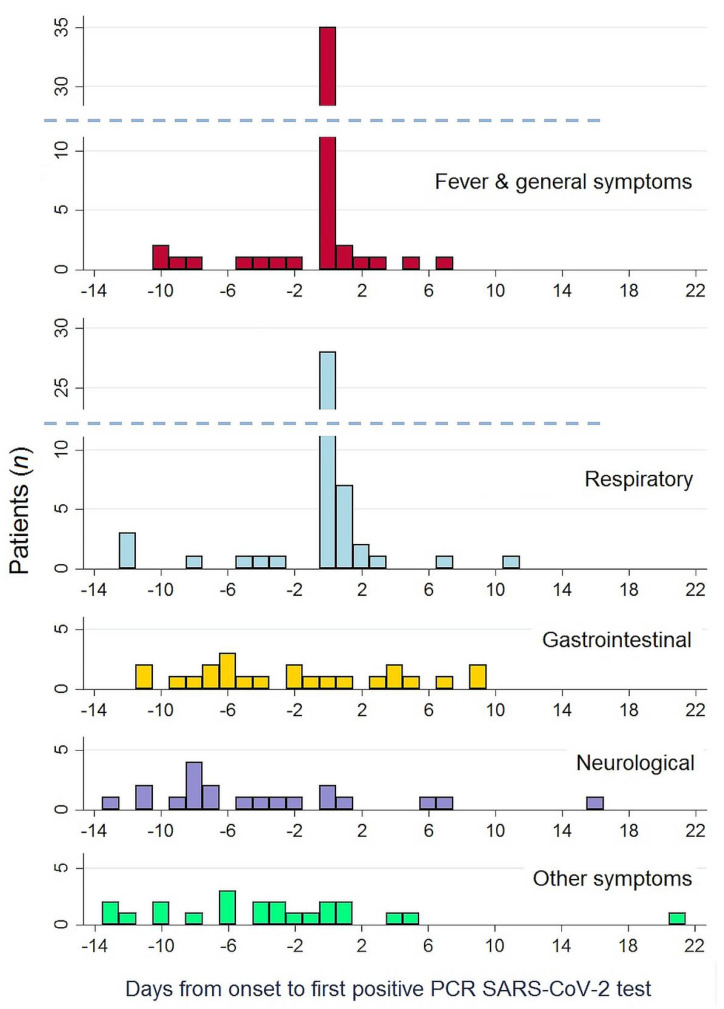
Temporal distribution of the onset of symptoms, by type.

**Table 1 jcm-10-02962-t001:** Signs and symptoms of COVID-19 in included patients.

	Confirmed COVID-19 Patients (*n* = 69)		PCR Negative Patients (*n* = 25)		
Variable	Patients *N* (%)	Onset of Symptom to COVID-19 Diagnostic (Days) Median (IQR)	Patients *N* (%)	Onset of Symptom to First PCR Test (Days)	*p*
Patients characteristics					
Age (mean, ±SD)	86 ± 6.4	-	82 ± 4.8	-	ns
Women	43 (75.4)	-	26 (70.3)	-	ns
ADL score (mean, ±SD)	3.25 ± 2.1	-	Not available	-	-
IADL score (mean, ±SD)	2.6 ± 2.5	-	Not available	-	-
Heart failure	26 (37.7)	-	8 (32.0)	-	ns
Coronary disease	16 (23.2)	-	0	-	0.008
Chronic lung disease	13 (18.8)	-	0	-	0.02
Cerebrovascular disease	19 (27.5)	-	7 (28.0)	-	ns
Dementia	32 (46.4)	-	15 (31.9)	-	ns
Cancer	6 (8.7)	-	6 (24.0)	-	0.049
Presenting symptoms					
No symptom at all	6 (8.7)	-	21 (84.0)	-	-
Fever, general symptoms	49 (71.0)	0 (0 to 0)	1 (4.0)	0	<0.001
Fever	46 (66.7)	0 (0 to 0)	1 (4.0)	0	<0.001
Malaise, asthenia	17 (24.6)	0 (-2 to 1)	0	-	0.048
Myalgia	2 (2.9)	−2.5 (−5 to 0)	1 (4.0)	0	ns
Respiratory	46 (66.7)	0 (0 to 1)	2 (8.0)	0 (both)	<0.001
Cough	32 (46.4)	0 (0 to 0)	1 (4.0)	0	<0.001
Oxygen desaturation	18 (26.1)	1 (−4 to 2)	0	-	0.002
Dyspnoea	14 (20.3)	0 (0 to 2)	0	-	<0.017
Abnormal lung sounds	14 (20.3)	1 (0 to 2)	1 (4.0)	0	ns
Respiratory failure	4 (5.8)	2 (1 to 3)	0	-	ns
Gastrointestinal	27 (39.1)	−5 (−9 to 3)	0	-	<0.001
Diarrhea	24 (34.8)	−3 (−7 to 4)	0	-	<0.001
Nausea, vomiting	8 (11.6)	−1 (−9 to 3)	0	-	ns
Abdominal pain	4 (5.8)	−11 (−17 to 11)	0	-	ns
Anorexia	2 (2.9)	−14 (both patients)	0	-	ns
Neurological	21 (30.4)	−8 (−11 to −3)	2 (8.0)	-	0.029
Delirium	17 (24.6)	−7 (−8 to −3)	2 (8.0)	−2 and 7	ns
New behavioural disorder	5 (7.2)	−11 (−13 to −3)	0	-	ns
Anosmia	1 (1.4)	0	0	-	ns
Other symptoms	20 (29.0)	−4 (−10 to 0)	3 (12.0)	-	ns
Fall, syncope	11 (15.9)	−4 (−12 to 1)	2 (8.0)	−2 and 21	ns
Heart failure decompensation	9 (13.0)	−3 (−8 to 0)	1 (4.0)	−6	ns
Outcome (at 30 days after inclusion)					
Dead	10 (14.5)	-	1 (4.0)	-	ns

ADL = Activities of daily living, IADL = Instrumental activities of daily living, IQR = Interquartile range, ns = Not significant.

## Data Availability

The data presented in this study are openly available in the Open Science Framework at DOI:10.17605/OSF.IO/WQC79, reference number WQC79.

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
