# Peer review of "Chronology of COVID-19 Symptoms in Very Old Patients: Study of a Hospital Outbreak"

_jcm, 2021, doi:10.3390/jcm10132962_

Round 1

Reviewer 1 Report

This is an original article regarding Chronology of COVID-19 Symptoms in Frail Older Patients.

The topic is promising but I have the following comments:

  • Not being experimental, the present study is retrospective even if the database is partially prospective. Please modify the text.
  • The authors should add the clinical implications of this manuscript
  • The discussion should be focused on the main topic and on comparison with literature data
  • English Language can be improved
  • References are not representative of the literature. At least: 1) COVIDSurg Collaborative. Mortality and pulmonary complications in patients undergoing surgery with perioperative SARS-CoV-2 infection: an international cohort study. Lancet. 2020 Jul 4;396(10243):27-38. doi: 10.1016/S0140-6736(20)31182-X. Epub 2020 May 29. Erratum in: Lancet. 2020 Jun 9;: PMID: 32479829; PMCID: PMC7259900; 2) COVIDSurg Collaborative. Delaying surgery for patients with a previous SARS-CoV-2 infection. Br J Surg. 2020 Nov;107(12):e601-e602. doi: 10.1002/bjs.12050. Epub 2020 Sep 25. PMID: 32974904; PMCID: PMC7537063; 3) Glasbey JC, Nepogodiev D, Simoes JFF, Omar O, Li E, Venn ML, Pgdme, Abou Chaar MK, Capizzi V, Chaudhry D, Desai A, Edwards JG, Evans JP, Fiore M, Videria JF, Ford SJ, Ganly I, Griffiths EA, Gujjuri RR, Kolias AG, Kaafarani HMA, Minaya-Bravo A, McKay SC, Mohan HM, Roberts KJ, San Miguel-Méndez C, Pockney P, Shaw R, Smart NJ, Stewart GD, Sundar Mrcog S, Vidya R, Bhangu AA; COVIDSurg Collaborative. Elective Cancer Surgery in COVID-19-Free Surgical Pathways During the SARS-CoV-2 Pandemic: An International, Multicenter, Comparative Cohort Study. J Clin Oncol. 2021 Jan 1;39(1):66-78. doi: 10.1200/JCO.20.01933. Epub 2020 Oct 6. PMID: 33021869; PMCID: PMC8189635.

Reviewer 2 Report

Many thanks for asking me to review this paper

The main point of interest is that you are looking at over 75 year old people. Most data in the 'elderly' starts from 60/65. I think that you need to emphasize this and I would include the age ranges of the other papers you quote looking at this area. I would even suggest that you add something along these lines in the title

You should also include the paper by Keevil etal, which considers a larger case number Geriatrics | Free Full-Text | Clinical Features, Inpatient Trajectories and Frailty in Older Inpatients with COVID-19: A Retrospective Observational Study (mdpi.com)

PLease also replace the word 'fragile' with 'frail' (pg 6, line 181) PLease include the reference paper of frailty and COVID-19 The effect of frailty on survival in patients with COVID-19 (COPE): a multicentre, European, observational cohort study - The Lancet Public Health

which highlights the increased mortality associated with COVID-19. This should be in the introduction and discussion. I would also reference the COVID-OLD study, which also showed that frailty predicts outcome in COVID-19 Frailty is associated with in-hospital mortality in older hospitalised COVID-19 patients in the Netherlands: the COVID-OLD study | Age and Ageing | Oxford Academic (oup.com)

You also have frail in the title. However, I could not see any assessment of frailty in the paper. This needs to be addressed as a limitation/or removed from the title

Thanks again

Round 2

Reviewer 1 Report

The authors improved the manuscript. English-language editing (professional) continues to be necessary

Reviewer 2 Report

Many thanks. You have made the required changes